# Record-High Efficiency Speckle Suppression in Multimode Fibers Using Cascaded Cylindrical Piezoelectric Ceramics

Ningning Yang [1,2], Zhicheng Li [2,3], Fanghao Li [1], Tingting Lang [4,*] and Xiaowei Guan [2,5,*]

1. Institute of Optoelectronic Technology, China Jiliang University, Hangzhou 310018, China; p21040854124@cjlu.edu.cn (N.Y.); lifanghao@cjlu.edu.cn (F.L.)
2. Jiaxing Key Laboratory of Photonic Sensing & Intelligent Imaging, Intelligent Optics & Photonics Research Center, Jiaxing Research Institute, Zhejiang University, Jiaxing 314000, China; lizhicheng@jrizju.com
3. Jiaxing Langtong Optoelectronics Technology Co., Ltd., Jiaxing 314000, China
4. School of Information and Electronic Engineering, Zhejiang University of Science and Technology, Hangzhou 310023, China
5. Centre for Optical and Electromagnetic Research, College of Optical Science and Engineering, Zhejiang University, Hangzhou 310058, China
* Correspondence: langtingting@zust.edu.cn (T.L.); guanxw@zju.edu.cn (X.G.)

**Abstract:** We present a technique that utilizes cascaded resonant cylindrical piezoelectric ceramics and multimode optical fibers wound around them to effectively mitigate laser speckle. By precisely driving the ceramics at their resonant frequencies and inducing comprehensive mode scrambling within the multimode fiber, we achieve a remarkable speckle suppression efficiency of up to 94%. To the best of our knowledge, this sets a new benchmark among various methods aimed at suppressing the speckle of a coherent light. Our study thoroughly explores variables influencing efficiency, including the cascading number of piezoelectric ceramics, driving voltage, fiber core diameter, and more. This method has significant promise for diverse applications that require efficient and fast control of speckle contrast.

**Keywords:** laser speckle suppression; piezoelectric ceramics; multimode fiber

## 1. Introduction

Laser technology plays a pivotal role in the information age, with applications spanning projection display, holography, imaging, microscopy, interferometry, tomography, and information processing, etc. This wide-ranging utilization is primarily due to lasers' narrow emission spectrum and high coherence which, conversely, provoke a negative effect known as speckle. Speckle manifests as a granular pattern created by the mutual interference of highly coherent light when it transits through a scattering media or is reflected from a scattering surface. Comprehensive details on speckle theory are readily available in various publications in the literature [1–3]. Its detrimental impact has many aspects, for instance, lowering image quality [4] or resulting in headaches in observers [5].

Numerous endeavors have contributed to suppressing speckle, primarily consisting of static and dynamic approaches. Static speckle suppression approaches involve utilizing wide-spectrum light sources to disrupt the spatial coherence of a laser [6] or employing fiber bundles of varying lengths [7] or refractive optical elements to disturb the temporal coherence of a laser [8]. Dynamic speckle suppression approaches involve the dynamic generation of uncorrelated speckle patterns within the integration time of the detector or eye and, consequently, suppress speckle through time averaging [2]. So far, a variety of dynamic techniques for mitigating laser speckle have been introduced. These methods include oscillating the projection screen [9], swiftly altering the lens focus [10], rotating a ball–mirror diffuser system [11], applying angle diversity to the beam by quickly vibrating piezoelectric benders [12], or vibrating multimode fibers [13–21].

Over the past two decades, and even before, dynamic speckle suppression involving multimode fibers has earned considerable interest. On one hand, multimode fibers exhibit low transmission losses, large core diameters, high numerical apertures (NAs), and the ability to handle high power. These characteristics render them popular in applications like spectrometers [22], endoscopy [23], holography [24], high-power light delivery [25], and even light detection and ranging (LiDAR) systems [20,21]. On the other hand, in these applications, speckle is readily induced due to the mutual interference between different modes in the fiber, subsequently leading to adverse effects.

While the implementation of static multimode fibers can help reduce a source's dynamic speckle [26] by carefully determining suitable fiber lengths, it is more beneficial to utilize dynamic methods. This is primarily due to the speckle's heightened sensitivity to external disruptions within the multimode fiber [27]. Moreover, dynamic strategies can help suppress speckle caused by the multimode fiber itself. Micro vibrators [20,21], a voice coil motor/attenuator [17,24], and even a fan [28] have been utilized to vibrate multimode fibers for speckle suppression. However, the drawback of these methods lies in their low frequencies, which range from tens to hundreds of Hz. Multimode fibers have also been attached to a piezoelectric plate [16,19] or wound around a cylindrical piezoelectric ceramic [17,24] to accomplish speckle suppression. However, in previous studies in which only one piezoelectric ceramic was used, the suppression efficiency was limited; even though the vibrating frequency could exceed 20 kHz, the suppression efficiency under a driving voltage of 20 V was only 47.83% [15], and it was 82.2% under a driving voltage of 10 V [16]. It is important to note that in this context, suppression efficiency represents the rate of change in speckle contrast when the vibrator is activated.

In this study, we employed cascaded cylindrical piezoelectric ceramics to vibrate a multimode fiber, aiming to amplify suppression efficiency. The cascading technique allows for a reduction in the number of windings around each ceramic, thereby facilitating fiber winding. Moreover, this scheme significantly reduces the driving voltage. Our experiments revealed that the speckle contrast could be dropped from an initial value of 0.65383 to 0.06719 at a driving voltage of 7 V, yielding a speckle reduction of an order of magnitude. The optimal speckle contrast reached was 0.0405, yielding a suppression efficiency of 94% at a driving voltage of 24 V, with operation frequencies exceeding 25 kHz. As far as the authors know, this is the highest suppression efficiency achieved to date. Additionally, comprehensive comparisons of the effects of fiber dimensions and the number of cascaded ceramics are also given in context.

## 2. Theory and Experimental Setup

### 2.1. Theory

Speckle contrast, denoted as $C$, serves as a common metric for quantifying speckle, expressed by Equation [2]:

$$C = \frac{\left(\langle I^2 \rangle - \langle I \rangle^2\right)^{1/2}}{\langle I \rangle}. \tag{1}$$

Here, $I$ represents the light intensity at a point on the investigated speckle profile, while $\langle I \rangle$ is the average intensity across the entire profile. Piezoelectric ceramics exhibit an elongation of electric dipole moments when an electric field is applied and thus deform through the piezoelectric effect [29]. Utilized as vibration carriers in speckle suppression, these ceramics offer high dynamics, a broad motion range, and a robust load capacity, resulting in excellent performances. The inherent frequency of a piezoelectric ceramic depends on intrinsic properties like its mass, shape, and piezoelectric constant. When the frequency of the employed sinusoidal waveform signal to the piezoelectric ceramic is aligned with the inherent frequency of the ceramic, resonating manifests during the vibration [30]. In the context of this study, the inherent resonating frequency of the piezoelectric ceramics was harnessed to mitigate speckle effects by inducing vibration in the optical fiber.

When a coherent laser beam is coupled with a multimode fiber, phase delays occur between the guided modes in the fiber, leading to interference between different modes and the formation of speckle patterns at the fiber's output. Vibrating the fiber disturbs the guided modes, causing random fluctuations in the phase delays between the modes and thus introducing decorrelation between these modes. Averaging the correlations between theses modes could also produce speckle contrast, mathematically expressed as follows [31]:

$$C^2(R,z) = \frac{\sum\sum_{m \neq n} I_m(R)I_n(R)|\gamma(\tau_{mn}(z))|^2}{\sum_m \sum_n I_m(R)I_n(R)} \tag{2}$$

Here, '$R$' denotes the position vector at the exit face of the fiber waveguide, '| |' signifies the modulus, '$\gamma$' symbolizes the temporal coherence of the exciting source field, and '$\tau_{mn}(z)$' represents the group delay time difference between mode $m$ and mode $n$ over the fiber waveguide length $z$. The equation suggests that lowering the temporal coherence '$\gamma$' will lead to a substantial decrease in speckle contrast. Using vibration in a multimode fiber can cause mode scrambling in the fiber, which significantly diminishes temporal coherence. At the same time, extending the vibrating fiber length can amplify the effect of mode scrambling. However, when few ceramics are used, the fiber needs to be wound many times to ensure sufficient vibrating fiber length. This requirement can complicate the winding process and increase the ceramics' impedance. As an example, our experiments showed that ten turns of fiber wound on a single piezoelectric ceramic increased the impedance by 327 mΩ, which then rose to 1166 mΩ for 40 turns. Therefore, implementing a strategy utilizing cascaded piezoelectric ceramics is anticipated to not only secure a longer disturbance length but also reduce the impedance burden for a single piezoelectric ceramic, ensuring lower driving voltages.

### 2.2. Experimental Setup

Figure 1 shows a schematic diagram of the experimental setup for the speckle suppression method proposed in this work. In our experiment, we used a 632.8 nm helium–neon laser as the light source from which the light was coupled into a step-index multimode fiber with a numerical aperture (NA) of 0.22. This multimode optical fiber was wound tightly around one or more cylindrical piezoelectric ceramics. Afterwards, the light egressed from the wound fiber and passed through a fiber collimator with a focal length of 10 mm to illuminate a charge-coupled device (CCD). The speckle profiles were extracted from the CCD images and mathematically processed to calculate the speckle contrast. The exposure time of the CCD camera was set to 50 ms, mimicking the integration time of the human eye. A signal generator was used to generate a sinusoidal signal, and the matching frequency was selected based on the measured frequency of the measured minimum impedance point. The output signal was amplified by 2× and applied to the cylindrical piezoelectric ceramics.

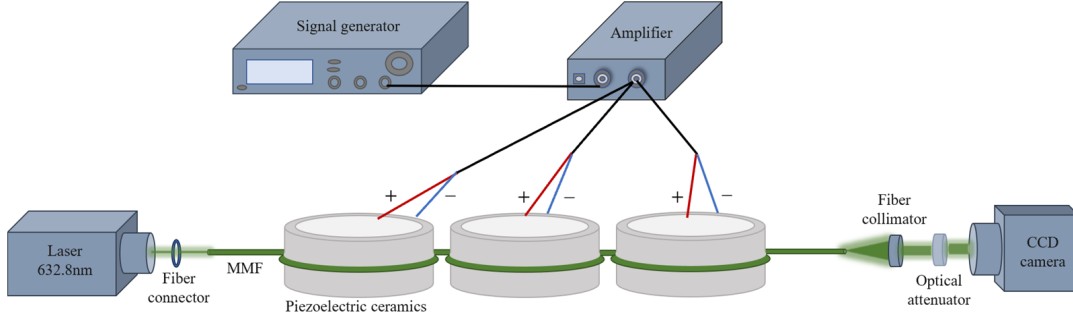

**Figure 1.** Schematic diagram of the experimental setup for laser speckle suppression using cascaded piezoelectric ceramics and a multimode fiber.

The resonant frequency of the cylindrical piezoelectric ceramics, as depicted in Figure 1, was determined through an impedance analysis using an impedance analyzer. The positive and negative poles were connected to the inner and outer sidewalls of the cylindrical piezoelectric ceramics. After the driving voltage of the sinusoidal waveform was applied, the impedances of the cylindrical piezoelectric ceramics at different frequencies were recorded. Here, we measured the impedance curves of three ceramics with different dimensional parameters, which are depicted in Figure 2 and identified as ch1, ch2, and ch3. The dimensional parameters are listed in Table 1. The frequency of the lowest impedance point for ch1 is 30.06 KHz, for ch2 it is 26.43 KHz, and for ch3 it is 25.54 KHz.

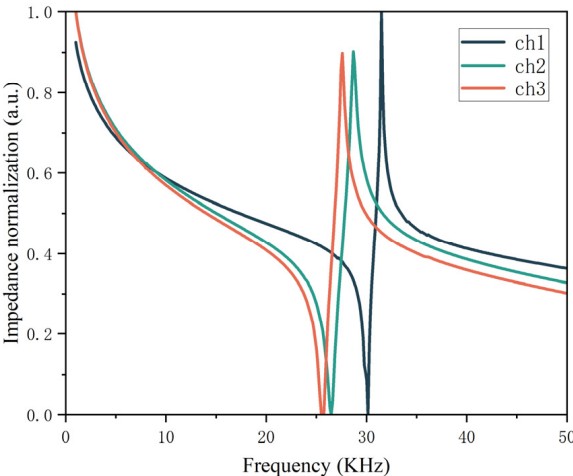

**Figure 2.** Measured impedance–frequency curves of three cylindrical piezoelectric ceramics with different dimensions.

**Table 1.** Related parameters of the used cylindrical piezoelectric ceramics.

| Parameter | ch1 | ch2 | ch3 |
|---|---|---|---|
| Resonant frequency (KHz) | 30.06 | 26.43 | 25.54 |
| Outer diameter (mm) | 38 | 38 | 38 |
| Thickness (mm) | $2.01 \pm 0.1$ | $2.66 \pm 0.1$ | $2.66 \pm 0.1$ |
| Height (mm) | 20 | 13 | 13 |

## 3. Results and Discussion

### 3.1. Effect of the Number of Piezoelectric Ceramics on Speckle Suppression Efficiency

To assess the impact of the number of piezoelectric ceramics on speckle suppression, three types of multimode fibers with core diameters of 125 μm, 200 μm, and 400 μm were used, and their measured speckle contrast values are shown in Figure 3a, 3b, and 3c, respectively. The experiments involved applying voltage to a single piezoelectric ceramic, two piezoelectric ceramics simultaneously, and three piezoelectric ceramics simultaneously for verification. Results across the three multimode fibers generally exhibit consistency. With an increasing driving voltage, the speckle contrast displays a saturated trend since the elastic limit of the ceramics is approached. The figures illustrate that speckle suppression is least effective when voltage is applied to a single piezoelectric ceramic. Applying a driving voltage to two ceramics results in less efficient suppression compared to cases involving three ceramics when the applied voltage is lower than 10 V. However, beyond 10 V, the speckle suppression efficiencies are almost the same for the cases involving two ceramics and those involving three. The results exhibited differences for the case involving a fiber with a 400 μm core diameter. Nevertheless, it is observed that when more ceramics are cascaded, a higher efficiency or lower driving voltage can be achieved. Meanwhile, when the applied voltage surpasses 20 V, the suppression contrast in each case drops below 0.1 for the cascade scheme.

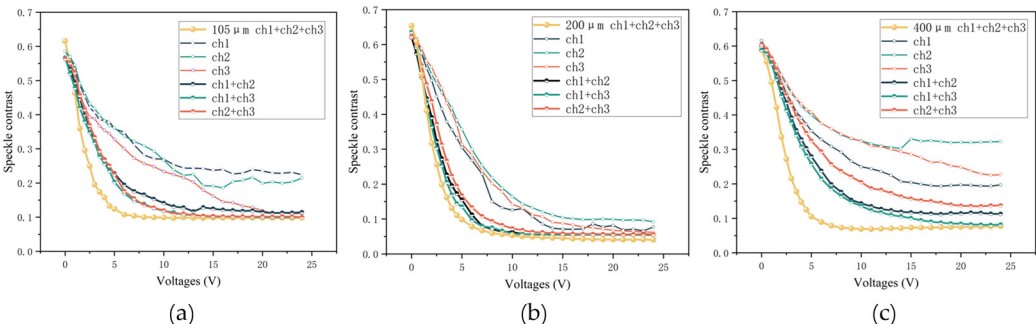

**Figure 3.** Measured speckle contrast values with respect to the applied voltage with a variable number of piezoelectric ceramics when the multimode fiber core diameters are 105 μm (**a**), 200 μm (**b**), and 400 μm (**c**).

*3.2. The Effect of the Number of Winding Rounds of Multimode Fibers on the Speckle Suppression Efficiency*

In the subsequent phase of the study, the impact of different numbers of winding rounds of multimode fibers on speckle suppression was investigated. The three cascaded ceramics were used in all experiments, and the driving voltage was simultaneously increased from 0 V to 24 V. This experiment explored the effects of winding the fiber once, three times, and five times. Figure 4a–c correspond to three different core diameters of the multimode fibers, 105 μm, 200 μm, and 400 μm, respectively. Across all samples, increasing the driving voltage led to a decrease in overall speckle contrast.

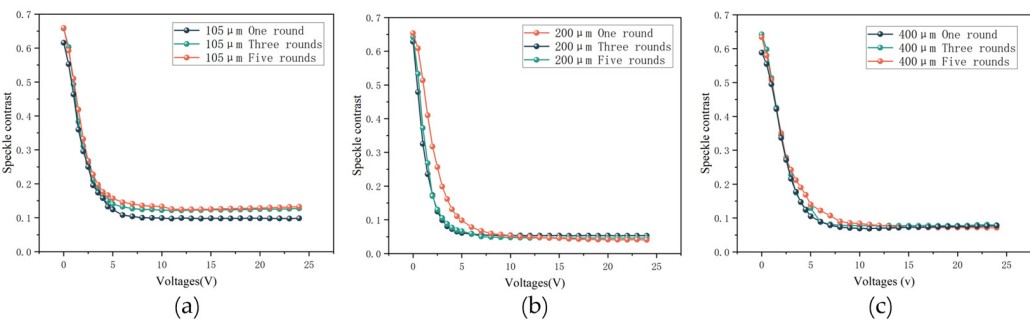

**Figure 4.** Measured speckle contrast with respect to the applied voltage under variable numbers of fibers wound on piezoelectric ceramics when the multimode fiber core diameters are 105 μm (**a**), 200 μm (**b**), and 400 μm (**c**).

In Figure 4a, for the case using the multimode fiber with a 105 μm core diameter, the curves for winding three and five times nearly overlap, exhibiting a saturated trend after the voltage is larger than 10 V. The lowest speckle contrast can be reduced to around 0.13. However, winding the fiber once results in a better reduction, with the lowest speckle contrast decreasing from 0.616 to around 0.098, achieving an 84% speckle suppression efficiency. As shown in Figure 4b, when using a fiber with a 200 μm core diameter, the three winding numbers result in final speckle contrasts of 0.0405, 0.05285, and 0.04589, corresponding to speckle suppression efficiencies of 94%, 92%, and 93%, respectively. In Figure 4c, for the experiment involving a 400 μm core diameter fiber, the three winding numbers show similar effects, with final speckle contrasts decreasing from 0.58834, 0.64232, and 0.63375 to 0.07731, 0.07889, and 0.07137, respectively, and corresponding speckle suppression efficiencies of 87%, 88%, and 87%. Our findings indicate that the number of fiber windings does not greatly influence the final speckle suppression efficiency. However, a more noteworthy observation was that decreasing the number of windings directly helps to reduce the driving voltage. This outcome aligns with the understanding that fewer windings can effectively curtail excess electrical impedance. Essentially, a strategic

reduction in winding numbers offers utility in managing operational voltage, thereby potentially enhancing the overall efficiency of the system.

### 3.3. Effect of Core Dimension of Multimode Fiber on Speckle Suppression Efficiency

In order to closely investigate how fiber dimensions influence the speckle suppression efficiency, we summarize the speckle contrast diagram for measurements involving different fiber core diameters but the same single winding round, as depicted in Figure 5. The measured results indicate that up to a driving voltage of 5 V, the speckle contrast reduction rate is similar for all three core diameter fibers. However, beyond 5 V, their performances diverge, with the 200 μm core diameter multimode fiber demonstrating the most effective suppression. This phenomenon may be linked to the unique characteristics of the fiber with a 200 μm core diameter, 358 mm vibrating length, and 10 m total length. According to Equation (2), the resulting mix of eigenmode numbers and their temporal decorrelation can enable peak speckle suppression. Nonetheless, vibration proves to be the dominant factor, given that the speckle contrasts for all tested fibers invariably converge to a value less than 0.12. This suggests that the strategic use of vibration, regardless of fiber specifics, can successfully suppress speckles.

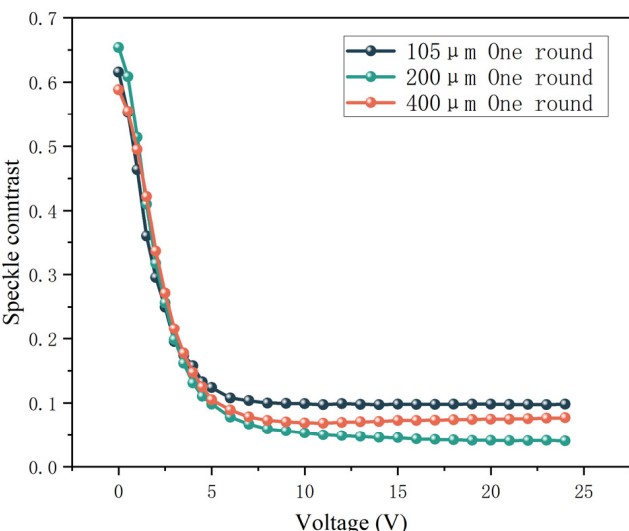

**Figure 5.** Measured speckle contrast with respect to the applied voltage using multimode fibers with different core diameters. Here, the experiments involved only one round of winding fiber on each of the three cylindrical piezoelectric ceramics.

We also extracted the spot profile images of the laser beam output from the fiber, as depicted in Figure 6a–c, for the measurements involving multimode fibers with 105 μm, 200 μm, and 400 μm core diameters, respectively. Note that all have one winding round of fiber. Both the greyscale images directly captured by the CCD and the transformed 3D images are shown, with the left side illustrating the case before voltage application and the right side illustrating the case after the application of 24 V.

As illustrated in the figure, an increase in the core diameter of the multi-mode fiber results in denser speckles which can be reflected in both the greyscale images and the 3D images. This is reasonable since a larger multimode fiber could transmit more modes and thus make the interference patterns denser. After adding a 24 V driving voltage, the speckle image for the 105 μm fiber still exhibits some blurry halos, and the grayscale image reveals surface roughness. Meanwhile, the speckle image of the 400 μm fiber shows slightly improved results. However, there is still some distinction in the intensity between the edges and the central region, evident in the grayscale image where the central region has slightly higher grayscale values than the corners. The best results are found in the measurements

using the 200 μm fiber, which demonstrate a more uniform overall distribution, with only some diffraction patterns coming from the lens.

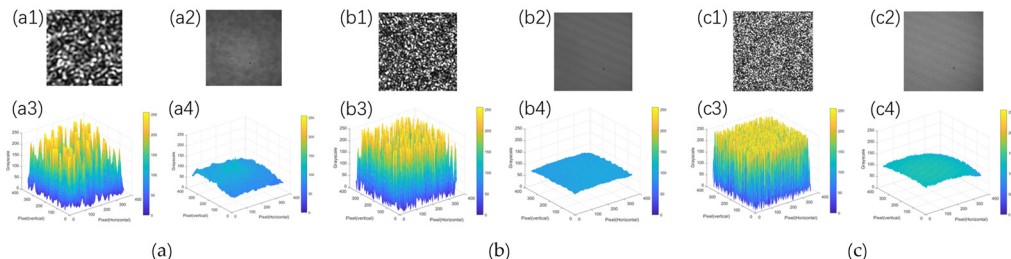

**Figure 6.** Comparison of the laser beam spot profiles for measurements involving multimode fibers with core diameters of 105 μm (**a**), 200 μm (**b**), and 400 μm (**c**). For each subfigure, the left column represents the measurement before voltage application, i.e., 1 and 3 in the figure; while the right one represents that after a voltage application of 24 V, i.e., 2 and 4 in the figure. Greyscale images were extracted directly for the CCD, and the transformed 3D intensity profiles are both exhibited.

To better demonstrate the superiority of the proposed speckle suppression method, Table 2 compares the speckle suppression efficiency and driving voltage of the method proposed in this study with other dynamic speckle suppression methods previously reported, whether multimode fibers were used or not. Here, the suppression efficiency signifies the change rate of the speckle contrast before and after the vibrator is activated. The higher the speckle suppression efficiency, the more effective the speckle suppression method. Generally, the methods involving no multimode fibers [32–36] exhibit lower suppression efficiencies. So do some of the methods using multimode fibers but no piezoelectric ceramics [28,37]. For methods using piezoelectric ceramics, whether they possess a cylindrical or a plate-like shape [15,16,19], the suppression efficiencies are smaller than those of the proposed method involving cascaded vibrating piezoelectric ceramics. For reference [15] in particular, although the method of using a cylindrical piezoelectric ceramic to vibrate the multimode fiber was also adopted, only one such ceramic was used. Thus, many turns, e.g., 20 in that work, of the multimode fiber had to be wound around the ceramic to achieve enough mode scrambling, resulting in a larger impedance and consequently a larger driving voltage. In contrast, this work innovatively introduces a cascading scheme in which only one winding turn of the fiber is needed, keeping a low impedance and driving voltage while securing high-efficiency speckle suppression. This is a further step toward a suppression-free laser beam for broad practical applications which is not limited to the advanced display.

**Table 2.** Comparison of speckle suppression efficiency using different methods.

| Reference | Driving Voltage (V) | Speckle Suppression Efficiency | Method |
|-----------|--------------------|-------------------------------|--------|
| [32] | 150 | 55% | Electroactive de-speckle diffuser |
| [33] | -- | 72.727% | Rotate diffuser |
| [34] | -- | 84.86% | Sports gum solution |
| [35] | -- | 68% | Replacement diffuser |
| [36] | 45–110 | 69.49% | Adjustable metasurface structure |
| [28] | -- | 72.45% | Multimode fiber vibrated by a fan |
| [37] | -- | 72.77% | Optical fiber vibrated by a voice coil motor |
| [19] | 150 | 81.395% | Multimode fiber vibrated by a piezoelectric plate |
| [15] | 20 | 47.83% | Multimode fiber vibrated by only one cylindrical piezoelectric ceramic |

**Table 2.** *Cont.*

| Reference | Driving Voltage (V) | Speckle Suppression Efficiency | Method |
|:---:|:---:|:---:|:---:|
| [16] | 10 | 82.2% | Multimode fiber vibrated by a piezoelectric plate |
| This work | 7 | 89.72% | Multimode fiber vibrated by cascaded cylindrical piezoelectric ceramics |
| This work | 24 | 94% | |

### 4. Conclusions

In this study, we proposed a method of using cascaded cylindrical piezoelectric ceramics to vibrate multimode fibers wound on the ceramics to tremendously suppress laser beam speckle or the speckle generated in the multimode fiber itself. The resonance effect was utilized by matching the frequency of the driving waveform to the inherent frequency of the piezoelectric ceramics, enhancing the stretch of the fibers and thus the mode scrambling of the light transmitting in the fiber. By employing the resonant and cascaded scheme, we found this dynamic speckle suppression method to be highly efficient, requiring only a small driving voltage. The speckle contrast decreased by one order of magnitude at a driving voltage of only 7 V and could be decreased down to 0.0405 at a driving voltage of 24 V, resulting in a high speckle suppression efficiency of 94%. Regarding numerous applications such as imaging, tomography, photolithography, and optical radar/metrology, which require speckle-free laser beam spots, we believe our proposal and verification of this speckle suppression method is a significant promotion beneficial to these applications.

**Author Contributions:** Conceptualization, Z.L., N.Y., X.G. and T.L.; methodology, Z.L. and N.Y.; software, N.Y.; validation, N.Y., Z.L. and X.G.; formal analysis, N.Y., Z.L. and X.G.; investigation, N.Y. and Z.L.; resources, X.G., T.L. and Z.L.; data curation, N.Y.; writing—original draft preparation, N.Y. and X.G.; writing—review and editing, X.G., N.Y., T.L. and F.L.; visualization, N.Y.; supervision, Z.L., T.L., F.L. and X.G.; project administration, T.L., X.G., Z.L. and F.L.; funding acquisition, T.L., X.G. and F.L. All authors have read and agreed to the published version of the manuscript.

**Funding:** This research was funded by National Natural Science Foundation of China (62205293, 52105595); the Zhejiang Provincial Natural Science Foundation of China (ZCLZ24F0501, LDT23F05012-F05, LDT23F05011F05, LTGS24E050002); Fundamental Research Funds for the Provincial Universities of Zhejiang (2020YW08).

**Institutional Review Board Statement:** Not applicable.

**Informed Consent Statement:** Not applicable.

**Data Availability Statement:** The dataset for this paper is provided.

**Acknowledgments:** We thank all the individuals and institutions that provided support for this research.

**Conflicts of Interest:** Author Zhicheng Li is employed by the company Jiaxing Langtong Optoelectronics Technology Co., Ltd. The remaining authors declare that the research was conducted in the absence of any commercial or financial relationships that could be construed as a potential conflict of interest.

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
