# Peer review of "Record-High Efficiency Speckle Suppression in Multimode Fibers Using Cascaded Cylindrical Piezoelectric Ceramics"

_photonics, doi:10.3390/photonics11030234_

Round 1

Reviewer 1 Report (Previous Reviewer 1)

Comments and Suggestions for Authors

I recommend the paper for publication in Photonics

Author Response

Reviewer 2 Report (Previous Reviewer 2)

Comments and Suggestions for Authors

The present paper reports about a method of speckle suppression by using vibration of multimode fiber wrapped around cascaded cylindrical piezoelectric resonator vibrator. However I cannot see the novelty of the paper when compared to reference 15 in which exactly the same idea was applied: “Efficient acousto-optic control of speckle contrast at the output of multimode fibers (MMFs) using a cylindrical piezoelectric transducer (PZT) vibrating in the radial direction. Ha, W.; Lee, K.; Jung, Y.; Kim, J.K.; Oh, K. Acousto-optic control of speckle contrast in multimode fibers with a cylindrical piezoelectric transducer oscillating in the radial direction. Opt. Express 2009, 17, 17536-17546”

There is no any discussion to demonstrate what is new in the present paper.

Comments on the Quality of English Language

Some corrections are needed

Round 2

Reviewer 2 Report (Previous Reviewer 2)

Comments and Suggestions for Authors

The authors properly corrected and improved, therefore i suggest acceptance

Comments on the Quality of English Language

minor points to be corrected

This manuscript is a resubmission of an earlier submission. The following is a list of the peer review reports and author responses from that submission.

Round 1

Reviewer 1 Report

Comments and Suggestions for Authors

A paper reports a method of speckle suppression by using vibration of multimode fiber wrapped around piezoelectric resonator vibrator. It is assumed that main application of the method is laser displays and projectors. In laser displays and projectors, one of the main problems is subjective speckle which arise in eye because of interference of scattering light from screen. However, the paper reports about suppression of objective speckle  that arise due to interference of modes of multimode fiber. Even more, since after fiber light passes through a collimated lens it is not pure objective  speckle at far end of the fiber since a different photodiodes  are illuminated by light of different modes due to different angles of their propagation. It is clear that statistics of objective speckle after multimode fiber is different from statistics of subjective speckle (Lapchuk at al. “Investigation of speckle suppression beyond human eye sensitivity by using a passive multimode fiber and a multimode fiber bundle” Optics Express 28 (5), 6820-6834, (2020 If to use an optical scheme propose in the paper for screen or display illumination it will result to a small speckle suppression effect since a one screen place will be illuminated only by coherent light that incident on screen at one angle, and therefore, there is not any angle diversity.

 For speckle suppression mechanism analysis is important to know an input NA of laser beam and NA of multimode fiber, however there is not information about these parameters in the paper.

A reference list should be improved to include the papers with most important results in the speckle suppression field.

A paper text also needs significant improvement:

 Line 49-50: a definition “Speckle contrast is defined as the ratio of the standard deviation of speckle intensity to the average speckle intensity”  is not correct.

Line 104-105: reference on [21] for Eq. (2) is not correct. Proper reference is “Hlubina JOURNAL OF MODERN OPTICS, 1994, VOL. 41, NO . 5, 1 001 -1014”

Line 108:  “complexity of the temporal coherence of the laser source”?

Finaly, in spite of many shortcomings, the experimental results about objective speckle suppression are interesting, therefore I recommend the paper major revision

Reviewer 2 Report

Comments and Suggestions for Authors

The idea about laser speckle suppression using piezoelectric ceramic resonance and multimode fiber is no new. This idea was already published by

Lin, Y., Fujimaki, Y. and Taniguchi, H. (2014) Reduction of Speckle Contrast in Multimode Fibers Using Piezoelectric Vibrator. Proceedings of SPIE “Laser Resonators, Microresonators, and Beam Control XVI”, 8960, 89601S-1-89601S-7.

Therefore, the results published do not introduce any novelty. Moreover, there is no any discussion or comparison between theory and experiments. The theory is incomplete.

Comments on the Quality of English Language

needs corrections
